# The Contribution of Pharmacists and Pharmacy Technicians to Person-Centred Care within a Medicine’s Optimisation in Care Homes Service: A Qualitative Evaluation

**DOI:** 10.3390/pharmacy9010034

**Published:** 2021-02-10

**Authors:** Sally Fowler Davis, Rachel Cholerton, Louise Freeman-Parry, Jo Tsoneva

**Affiliations:** 1College of Health Wellbeing and Life Sciences, Sheffield Business School, Sheffield Hallam University, Sheffield S1 1WB, UK; R.CHOLERTON@shu.ac.uk (R.C.); l.freeman-parry@shu.ac.uk (L.F.-P.); 2Sheffield NHS Clinical Commissioning Group, Sheffield S9 4EU, UK; jo.tsoneva@nhs.net

**Keywords:** medicines optimisation, person centred care, qualitative, framework analysis, evaluation

## Abstract

Pharmacists and pharmacy technicians seek to improve person-centred care. Improvements to systems for care homes seeks to reduce medicines waste and inefficiency, particularly through supporting care home staff, to enhance safer administration of medicines. A complex evaluation used qualitative design and utilised narrative enquiry, and team members and key stakeholders were interviewed. Framework analysis was used, aligning findings to a person-centred care framework for older people. The Medicines Optimisation in Care Homes (MOCH) team brokered improvement practices across care homes to enhance person-centred care. The framework analysis confirms that the team used ‘authentic attention’ in relation to the residents’ experiences and flexibility in relation to negotiating medication. The importance of transparency of processes and systems in medicines management is highlighted, alongside requirements for person-centred care to make explicit the reason for taking a medication, and the continuous discussion with a range of stakeholders about the continuing need for particular medications. The outcome of the evaluation includes insights into a new area of pharmacy practice in community, based on the skills, knowledge, and experience of pharmacists and pharmacy technicians working in the care home sector. Further study is needed into the efficacy and outcomes of medicines management interventions.

## 1. Introduction

Previous studies have examined the impact of deprescribing processes in residential care settings [1], where pharmacist-led medication reviews are reported to slightly decrease numbers of drugs prescribed and interventions could improve medication knowledge and adherence [2]. Some pharmacist-led medication review interventions have revealed that the cost of medications was reduced [3], although this had no effect on other quality indicators, such as hospital admission. There is evidence of significant medication errors in the care home context [4,5], and it is well recognised as being a challenging environment for introducing new practices [3,6]. The wider contribution of pharmacy services has been under-evaluated. Little is known or has been formally reported about whether quality of life of residents was improved in other ways through medication-related interventions, and what contribution was made to the care home experience by medicines management practices. 

The Kings Fund review [7] considered strategies to improve medicines management and polypharmacy including medication review, reconciliation, medicines management, and reduction of medication waste. It also recommended that all patients in care homes should have their medications reviewed by a pharmacist and that one person (possibly a pharmacist) should have overall responsibility for medicines used in the care home [7]. In the same year, the Royal Pharmaceutical Society published “Medicines Optimisation: Helping Patients make the Most of Medicines” [8], a guide for healthcare professionals in England to support patients to get the best outcomes from their medicines use. The document describes four guiding principles for medicines optimisation using a person-centred approach. Medicines optimisation has been defined as “a person–centred approach to safe and effective medicines use, to ensure people obtain the best possible outcomes from their medicines” [8]. 

National Institute for Health and Care Excellence (NICE) Guidance 2015 [9] was created for healthcare professionals to ensure that medicines provide the greatest possible benefit to patients. This report identified key priorities around medicines adherence, advocating that patients could be involved in decisions about their medicines and that healthcare professionals need to take account of patients’ needs, wishes, and preferences. Other key priorities included identifying, reporting, and learning from medicine related patient safety incidents, medicines reconciliation, and review and development of patient-centred medicines-related models of organisation and cross sector working. One of the recommendations (1.8.2.) was that organisations should involve a pharmacist with relevant clinical knowledge and skills when making strategic decisions about medicines use or developing care pathways that involve medicines use. NICE [9] also recommended research into cross sectional workings to identify models that improve clinical and cost effectiveness in relation to sub optimal prescription of medicines.

This evaluation project aimed to investigate the outcomes of the Medicines Optimisation in Care Homes (MOCH) team, with reference to the wider person-centred care outcomes achieved alongside the more traditional outcomes related to medicines and prescription. The MOCH team comprised of eight individuals working in pairs, involving a pharmacist and pharmacy technician. The team was funded as a pilot by NHS England and worked semi-autonomously with professional guidance from a Management Committee hosted by the NHS Sheffield Clinical Commissioning Group (CCG). As a new service, team structure was continuously being developed. The team worked flexibly across five sub regions with around 180 care homes, covered by five CCGs within the South Yorkshire & Bassetlaw Integrated Care System (ICS). The investigation sought to identify facets of person-centred care enabled by the team.

## 2. Methods 

Narrative inquiry [10] was used to underpin the methods of this evaluation. This approach was used due to the method allowing understanding of the individual’s story, and to “adopt a particular view of experience as phenomenon under study” [11]. The emerging and novel interventions in care homes were the phenomena that enabled person-centred care in relation to medicine optimisation and the administration of drugs. Nine in-depth interviews took place between March and April 2020. Ethical approval was provided by [SFD, Sheffield Hallam University] (Approval ID: ER22134371, March 2020). As this study was classed as a service evaluation, further approval from Health Regulatory Authority (HRA) was necessary. The study was sponsored by the NHS Sheffield CCG. As the research was commissioned with an aim to further best practice and provide a “theory of change” in medicines optimisation, managers of the service were involved in reviewing the study findings, but not interviewed. The paper is reported following the Consolidated Criteria for Reporting Qualitative Research (COREQ) [11].

The research team involved in this study were multidisciplinary and have experience in backgrounds related to the focus of this paper. SFD is an Assistant Professor in Organisation in health and care. LFP is a PhD and registered pharmacist in the UK. RC (the lead interviewer) is a postgraduate researcher currently completing their PhD, within the field of behavioural change, and has published previous qualitative research papers. JT is a pharmacy development manager within a clinical commissioning group in the NHS. All of the primary research team were female.

A semi-structured interview guide was used to direct interviews and answer questions pertaining to the evaluation of the MOCH intervention. Interview guide development was guided by key principles of narrative inquiry [10], focusing on an introduction to the participant’s role within the intervention, and then asking the participant to describe a specific situation or resident which was impacted by the MOCH intervention, and describe the process in which the situation developed. Further probing questions were asked if the participant asked for more detail, including information around the initial conditions and circumstances of the resident, what the process of the intervention involved for this specific patient, and questions surrounding the end result for the resident, care home, and wider organisation. The topic guide for the interview is presented in Appendix B.

Team members (MOCH pharmacists and pharmacy technicians) and stakeholders (care home managers and one manager from the CCG) were invited for interviews. Recruitment for the study was purposive, and potential participants were contacted via email to introduce the study aims and to provide a participant information sheet. All MOCH team members were invited for interview and other stakeholders invited included registered care home managers receiving the MOCH service, and those holding a senior management position at a local NHS Clinical Commissioning Group.

The interview schedule was prepared, and one member of the research team (RC) conducted all interviews, either in person (*n* = 4) or by telephone (*n* = 5). Eleven individuals were invited for an interview for the study and nine participants took part. Two individuals did not take part due to absence and change of role. No other people were present during interviews. RC had no prior relationship with any participants before briefing and interviewing those taking part in the study, and participants only knew of the interviewer through the study being conducted. RC, when briefing participants, made aware of her position as a researcher within the university, from a non-clinical background. Participants took part in one interview, during working hours, to suit their schedule. Before the interviews, the researcher sent a preparation document (the topic guide) to allow participants to prepare for a semi-structured interview (see Appendix B). The duration of interviews were between 26 and 53 min. Interviews were recorded and professionally transcribed. Every effort was made to anonymise narratives, and information was compiled removing any features of the care home environment or practitioner. Interviewer notes and reflections were noted separately on interview guide sheets as the interview took place. Due to the niche participant sample collected, and the onset of the COVID-19 pandemic and work constraints for the participants, the research team decided on data saturation at nine participants.

Framework analysis, defined as a method which systematically reduces the data to analyse it by case and by code [12] was used to analyse data from transcripts. Framework analysis was chosen by the researchers, as the method allowed researchers to compare and contrast participants’ experiences, whilst allowing for each participant’s experiences to remain connected to other aspects of their account, so to not lose context of individual views [13]). Two researchers read and independently coded the first three interview transcripts and then discussed codes. This method was used to compare participants’ experiences, whilst allowing for each participant’s experiences to remain connected to other aspects of their account, so to not lose context of individual views [13]. Framework analysis typically follows seven stages of analysis: transcription, familiarisation of the data, coding, developing a working analytical framework, applying the analytical framework, charting data into the framework matrix, and finally, interpreting the data. After discussion, McCormack’s (2003) “conceptual framework for person-centred practice with older people” [14] was deemed a close fit to the initial coding. The data was then charted manually, without the use of software, with some inductively developed additional subheadings relating to practitioner enablers, barriers, and outcomes (relating to the resident, practitioner, care home, and wider organisation) to allow for the data to be fully represented in the analysis. After charting the data, data were interpreted by the lead interviewer (RC) according to the framework [14]. These interpretations were then discussed amongst the evaluation team. Due to time constraints and the COVID-19 pandemic, member checking of the transcripts by participants were not issued. However, draft reporting enabled the participants and management to comment and reflect on the findings.

A synthesis of qualitative findings was derived from the framework analysis by way of providing an indicative analysis for each aspect of the MOCH intervention as a person-centred intervention (Appendix A).

## 3. Results

The MOCH team reported experiences in 28 care homes. There were 10 Residential and 16 Nursing Homes across the MOCH programme remit. Individual narrative accounts revealed specific outcomes in relation to person-centred care for older adults, against five themes related to the conceptual framework devised by McCormack [14]. 

### 3.1. Authentic Consciousness. 

This theme refers to a commitment from the pharmacist to consider the resident in the context of their lived experience and the meaning to their life in the care home. The MOCH team often stated the approach to assessing a resident and did so in a holistic manner. Many team members assessed past medication history and lifestyle history, as well as current factors affecting any symptoms. One team member stated that:

“we don’t just look at the medication, we’ll look at everything, including nutrition, fluid intake…you know, exercise if they’re getting that, any sort of issues they may have with like the dexterity or… if they’re able to take the medications themselves.”(*P8, MOCH team member*).

Notably, stakeholders also highlighted MOCH team members spoke to the resident to gather this information, with one stakeholder discussing that:

“it was literally a case of going through like the basics, like learning, um, her nutritional status, her past medical history, her allergies, her eating and drinking, um, and a lot of it actually came from her herself [the resident]”(*P6, stakeholder*).

The MOCH team members examined in thorough detail the medication reviews to understand the resident. In doing this, the pharmacists noticed previous poor communication related to previous rapid reviews and not fully understanding the resident’s needs:

“What I realised is the type of reviews that the pharmacist was doing with the senior nurse and they were spending a whole day going through the 50, 60 residents in that one day. It’s not a proper review. So, what we were able to do is do a full holistic and full medication review with all the other professionals”.(*P5*, *MOCH team member*).

### 3.2. Informed Flexibility

This theme comprises of the commitment by the carer to the information sharing about medication and the formalisation of a decision-making process about the personal use of medication. The perception in the team was that the whole multidisciplinary team were involved in the medicine optimisation process, “raising the awareness of medicines and condition management amongst the nurses, carers, and the family, so in other words it’s a holistic care.” (*P1, MOCH team member*). Additionally, the education of how different lifestyle factors may affect residents’ symptoms was also an important discussion point raised. For example:

“It was a case of sitting down with [resident] and looking at her [medication] regime that she was on, which had been initiated by the pharmacist as well, so it was explaining in a little bit more detail that that would help with any potential [symptoms]. We did some education around adequate hydration. We looked at, you know, her water intake over the day, and we explained the benefits”.(*P2*, *MOCH team member*).

The variation of knowledge regarding medicines was a barrier, when working with staff of different levels and education on certain medicines. One care home staff member said “I’m a nurse, I’ve had medications training, but I struggle… I don’t understand it in the way that they see it at all.” (*P6*, *stakeholder*)

### 3.3. Mutuality

The theme is defined as “the recognition of the others’ values as being of equal importance in decision making”, within McCormack’s framework of person-centred care in older people [14]. Within the MOCH context, discussions focused on the commitment to choose medications, acknowledging the complex meaning of the use of medication for the individual, their relatives, and the wider team. One MOCH team member explained the importance of having family input where the resident may not have the capacity to engage in conversations:

“Also the family, I want to get their views… so the nurse and the carers gave me the details about this resident since the admission to that care home, um, and how they were trying to manage the symptoms… then obviously I spoke to the family to get their viewpoint on how this stuff is managed, because care home feedback is one side of the coin, the family feedback is the other side of the coin, so it’s important to have both sides reviewed”(*P1*, *MOCH team member*).

Equally, being aware of the care home’s views on the MOCH intervention before starting resident medication reviews was important, to make sure values of that individual care home were taken into account. One MOCH team member discussed building rapport and trust was a vital part to establishing a relationship with the care home:

“I think whenever you start in a care home there’s always a bit of trepidation, a bit of anxiety from the care home staff about who are we, you know, are we the CQC (Care and Quality Commission), are we checking up on them, are we inspecting, are we, you know, judging them… so, initially whenever me and my colleague [name] went in a care home there is always that level of uncertainty about who we are and what we’re going to do, so it’s establishing that trust and building that rapport.”(*P3*, *MOCH team member*).

Conversely, some barriers to providing person-centred care included varied receptiveness of care home managers to the MOCH initiative. One MOCH team member highlighted low receptiveness from one care home manager, expressing reluctance to involve the team in the care home. They stated that:

“the care home manager was not keen for us to be there and, uh, didn’t really want us there, but, because the CCG had said, he was quite reluctant to let us in, so wanted to know when we were coming, what time we were coming, how long we were going to come for, what our job was”.(*P5, MOCH team member*).

### 3.4. Transparency

The theme refers to the commitment to the knowledge, skill, and management of dispensing processes in the home as a means of optimising medicines management and improving the quality of dispensing [14]. This involved communicating clearly with all parties involved in the resident’s care and depended on a range of skills and experience, including documentation regarding both medications and non-pharmacological interventions. Many participants discussed forming a constant line of communication with carers on any drug side effects or changes to resident medications:

“We do say once we come out of the care home you can always contact us, we’re always there if you’ve got any queries and we’ll signpost you to the correct person. So, it’s not a case that we go in, we do what we’re doing and then we come out and they’re left unsupported”.(*P8*, *MOCH team member*).

Additional follow ups with the resident and, where necessary, the family of the resident, were discussed amongst the team members. This was particularly noted with one team member, highlighting how unnecessary medications were avoided being dispensed, due to keeping a clear line of communication with the family of one resident. They stated that:

“Since we started withdrawing some drugs, then the patient did have some symptoms and family, because they were, you know, heavily involved and they participated really well…they contacted me, I could go back in again and put that extra reassurance into the system and have that extra discussion with the family”.(*P1, MOCH team member*).

However, before the MOCH team intervention, some care homes had medication processing systems which were not synchronised or not up to date, causing confusion and a lack of transparency amongst staff members. One stakeholder discussed that they were “finding some prescriptions that we were thinking hadn’t arrived or were missing from the cycle weren’t actually repeat prescriptions in the first place” (*P6*, *stakeholder*). This was echoed by the team members when entering care homes and assessing medication processing systems, with one team member stating that old order sheets were often slow to update, and “that is what they’ll use in then a few weeks’ time to reorder, not realising that things have been changed.” (*P2*, *MOCH team member*). An additional barrier to transparency was related to staff absence, if one person was absent from work, it had an effect on the quality of service the remaining team member could offer the care home:

“I had a lot of difficulty because my technician was off quite a bit, which had a real big impact, because there wasn’t that consistent…what we said is, we’re going to be looking at reviewing the policies, we’ll be looking at this, you know, so we’d sold it quite well… then, you know, uh, my technician was off, uh, for a little while as well. So, you know, you lose the quality kind of approach”.(*P5*, *team member*).

### 3.5. Negotiation

This refers to a commitment to resident participation in decisions about medication use and acting as an intermediary for the resident in managing care processes within the care home [14]. MOCH team members widely discussed the involvement of the resident as much as possible throughout the medicine optimisation process. One team member highlighted the importance of having resident-led conversation where possible:

“We go with maybe an idea of what we want to talk about and discuss, but right from the start the tone of the conversation you’ve got to get set by the resident, what they want and what, what they hope to get out of it and where they would like to finish up with at the end of the review or be on the journey to towards achieving”.(*P3*, *MOCH team member*).

However, some barriers to negotiating with the resident included whether the resident was able to respond to conversation. Some residents cared for in some situations were not compos mentis, or not responsive to MOCH team member engagement, with one team member reporting “I went and, uh, spoke to him he didn’t speak at all, he was just lying there focused and he didn’t even respond or anything.” (*P5*, *MOCH team member*).

Despite this, MOCH team members aimed to gain agreement from as many parties involved in the resident’s care as possible. The process included addressing resident or family concerns about any changes in medication, and involving the family and care home staff in a bigger capacity if the resident was not able to communicate with the MOCH team member:

“[The patient] also highlighted that they’d had leg ulcers before and they were really concerned about getting these leg ulcers again, because they’d gone through a long time of having their legs bandaged, um, and didn’t want to go back to that, so we recommended that they get some stockings fitted and we looked at introducing exercises.”(*P8*, *MOCH team member*).

### 3.6. Sympathetic Presence

This theme suggests a commitment to the personhood of the older adult in the care home and adapting methods and processes of deprescribing, and other dispensing advice, to accommodate a level of understanding, cognitive changes, and maximising options within the culture and life of the care home [14]. MOCH team members also recognised the importance of advocating for the residents’ wellbeing, in situations not wholly related to medication optimisation, where it was influencing their quality of life. One MOCH team member highlighted how the resident’s environment also affected her quality of life:

“It’s a two-floor care home and she was on a floor, with dementia patients, but she didn’t have dementia, so in terms of communicating with anybody she found it extremely difficult… so we explained to the care home manager that it wasn’t the most appropriate environment for her and actually it’s probably leading more to her, to her isolation”.(*P2*, *MOCH team member*).

One MOCH team member highlighted a previous medication change was overlooked, that could have had a positive effect on the resident, stating “it was actually initiated by the GP, but it was overlooked. It was overlooked as an intervention that could have had a better outcome for [resident]” (*P2*, *MOCH team member*).

### 3.7. Context of Care Environment

Lastly, this theme refers to an understanding of the whole system and the opportunities and barriers to practice within the team and within individual care homes, including, but not limited to, variations in power relationship between care providers, knowledge, skills, and quality processes within the particular context, and the physical space offered [14]. One underlying aim expressed by MOCH team members was giving residents the autonomy over their own health choices, which was noted by stakeholders, expressing “I think it gave them [residents], it’s given them that, opportunity to sort of have a bit of a what do they want to achieve in their health” (*P6*, *stakeholder*). 

Furthermore, one participant discussed the importance of noticing the small and simpler of tasks on the wellbeing of the residents, stating that “It can be the simplest of things that are often overlooked, but when you’ve got the luxury that we have of time in the care homes, then you can look at all that sort of stuff and it’s that really that I think is the most important.” (*P8*, *MOCH team member*).

## 4. Discussion

The study clearly reveals a commitment to person-centred care that seeks to balance the technical demands of medicines review and optimisation with the priorities of care home residents, their family carers, the care home managers, primary care providers, as well as other stakeholders. Clinical and therapeutic areas of activity included a holistic understanding of the resident experience and a careful understanding of a wide range of clinical and personal factors. This is a complex and diverse area of practice where knowledge of dementia, pain, and antipsychotic and cardiovascular medication is combined with the assessment and provision of the service and contextualized in relation to effective multidisciplinary working and care home staff training. The clinical reasoning associated with complex case management and medicines review requires knowledge, skills, and experience that are not well codified within pharmacy practice [15].

The qualitative data was combined and synthesised into a table (see Table 1) to reflect the different aspects of implementing the pilot MOCH services. These include: resources, inputs, activities, outputs, outcomes, and impact, and specific references to qualitative data are linked to each aspect to reflect the information shared by the management team and the interviewees, with content that demonstrates the regional response to the MOCH implementation. In this way, the evaluation reflects how the resources are deployed and shaped into a service specification that includes a range of activities. The interviews revealed how the activities, implemented at regional level as the MOCH, resulted in outcomes and an impact in relation to quality of care. 

The evaluation sought to enable the service provider to select and share the most important aspects of the service and the contribution to person-centred care outcomes within the pilot. This evaluation, therefore, serves to identify some of the professional competencies that contribute to care homes. As a number of core outcome sets (COS) have now been developed, there is perhaps a need to balance the technical, core contribution, (i.e., medication review, adverse drug events, prescribing errors, and associated costs) with practical care considerations. The measures of effectiveness for services are currently associated with optimising prescribing for older adults in care homes [16,17], and the practical knowledge and training needs of pharmacists in care homes may usefully include an accreditation process for the care home sector [18]. Physicians have previously emphasized the importance of effective communication, pharmacist access to the medical record, and a mutually trusting relationship as key attributes of a program [19], to integrate pharmacists’ roles on health care teams [20]. The Standards for pharmacy professionals from the General Pharmaceutical Council (pharmacyregulation.org) report sets out the vital role of delivering person-centred care that involves and enables people to make decisions about their health, safety, and wellbeing, and it is critical that professionals are able to fully appreciate and recognise the importance within their practice.

Further acknowledgement and education involving stakeholders may be helpful for practitioners working in the community and in the care home sector [15,21]. As indicated in the policy development and more recently supported within the Enhancing Health in Care Homes report (NHSE 2020), the presence of a pharmacist as a ‘trusted’ and knowledgeable professional, supports a key quality improvement indicator for care homes. The individual knowledge and confidence of the pharmacist and pharmacy technician will be an important factor, recognising the complex range of individual health and care needs of the resident population and the cross sector working practices. A limitation of the study was that data saturation could not be achieved, and adequate saturation was reached with nine participants but it was recognised that this could not fully describe the new practice, and so further study and practice reflection would be an important opportunity to describe pharmacy practice in the care home sector.

Medicines Optimisation is a recognised method of improving choice and opportunity for deprescribing for care home residents, and in the UK, there are teams of pharmacists and pharmacy technicians supporting the administration of medications within care homes. Pharmacists are contributing to person-centred care in care homes using technical knowledge and negotiation skills. The scale of this evaluation study limited the investigation to qualitative descriptions of the medicines optimisation team’s reported activity. Further research is needed to investigate the impact of person-centred care in the management of medicines and outcomes in relation to adherence and deprescribing.

## Figures and Tables

**Table 1 pharmacy-09-00034-t001:** MOCH intervention and person-centred evaluation.

Resources	Input	Activities	Outputs	Outcomes	Impact
The team comprises of three clinical pharmacists and five pharmacy technicians working across care homes in South Yorkshire and Bassetlaw	The remit is to work with care homes and with residents to provide medication reviews, with specific pharmacy advice to residents and primary care. To work with care homes on safer administration	-Medicines review -Deprescribing -Risk appraisal of polypharmacy -A holistic approach to patient care -Medicines reconciliation -Training and advice on administration and medicines management -Capability with ordering and supply	-Individual medication burden reduced -medicines management in the care home improved -Personalised outcomes related to quality of life improved understanding of medication and choice -care home competency	-Relational network and point of contact for medication management -Deprescribing with associated cost saving -Systems knowledge and expertise regarding pharmacy review and deprescribing -Individual health outcomes	-Increased quality of care, as indicated by safe use of medications -Quality regulation and assurance via CQC-Population health benefits
**Indicative narrative**	**Indicative narrative**	**Indicative narrative**	**Indicative narrative**	**Indicative narrative**	**Indicative narrative**
“We do say once we come out of the care home you can always contact us, we’re always there if you’ve got any queries and we’ll signpost you to the correct person. So, it’s not a case that we go in, we do what we’re doing and then we come out and they’re left unsupported” (*P8*, *MOCH team member*)	“a full holistic and full medication review with all the other professionals.” *(P5*, *MOCH team member*)	“we’ll look at everything, including nutrition, fluid intake… you know, exercise if they’re getting that, any sort of issues they may have with like the dexterity or… if they’re able to take the medications themselves” (*P8*, *MOCH team member*)	“We go with maybe an idea of what we want to talk about and discuss, but right from the start the tone of the conversation you’ve got to get set by the resident, what they want and what, what they hope to get out of it, and where they would like to finish up with at the end of the review or be on the journey to towards achieving.” (*P3*, *MOCH team member*)	It is just having that knowledge of who they [care home staff] need to go to, because before it was always, oh, we’ll speak to the GP, and then it sort of went, and then the GP was expected to, you know, ring the memory services or the falls clinic, or whoever, whereas now they know that they can speak to the pharmacist, they don’t have to speak to the GP, if, if they’re querying a medication they can speak to the pharmacy (*P7*, *Stakeholder*)	“wasn’t requesting a doctor’s visit on a daily basis, which she was doing before just out of fear” (*P8*, *MOCH team member*). We have seen a huge reduction in medication errors already… I’ve learnt so much about medicines myself that I feel more confident to tackle now.” (*P6*, *stakeholder*)

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
