# Peer review of "The Contribution of Pharmacists and Pharmacy Technicians to Person-Centred Care within a Medicine’s Optimisation in Care Homes Service: A Qualitative Evaluation"

_pharmacy, 2021, doi:10.3390/pharmacy9010034_

Round 1
Reviewer 1 Report
Overall Comments:
This is an interesting area of research where pharmacists and pharmacy technicians using person-centred care is seen to have the potential to improve medicines’ optimization. The manuscript frames person-centred care as a practice that can be applied within medicines’ optimization practice and subsequently, may improve patient quality of life. Hence the authors have looked to identify and evaluate the use of person-centred care within existing practice in a defined setting (care homes service), that is, a retrospective evaluation.
However, there are several important points that need to be described in detail or more detail to improve the manuscript.
Some (but not all) specific points are addressed in detail in the attached document.
Introduction General Comments:
The aim and relative objectives need to be made much clearer.
There is no clear justification for the choice of method.
Lines 30-31: Reference 1 is missing. It is incorrect for the referencing to start with Reference 2. Please correct.
Lines 48 -49: Patient-centred and person-centred are not interchangeable terms. They are two distinct types of practice. Please clarify the difference between the two and explain which of the two practices is being used in the manuscript.
Line 49 – where does the quote end? There is no quotation mark (inverted commas) to close the quote.
Comments on methods section:
There justification for choice of methodology should be clearer.
The methods section does not provide enough detail for the study to be replicated which is a general requirement of a methods description.
Lines 75-76 states that the “managers of the service were involved in reviewing the information for the study”. Why was this so and how could this have impacted the study.
The authors refer to having used the COREQ checklist in lines 76-77 which is appropriate. The COREQ checklist provides a “checklist of items that should be included in reports of qualitative research”. However, several COREQ items have not been reported. These missing items may influence what the participants felt comfortable in disclosing during the interviews and hence may have significantly impacted the results. These missing items should be reported to ensure the quality and rigor of the qualitative research such as sincerity, self-reflexivity, credibility, transparency, as listed (per domain and by item number) below. If the item has not been addressed then this should be explained in the methods section (in accordance with the chosen methodology) or addressed in a limitations section.
- Domain 1:
- 2 Credentials,
- 3 Occupation,
- 4 Gender,
- 5 Experience and training,
- 6 Relationship established,
- 7 Participant knowledge of the interviewer,
- 8 interviewer characteristics,
- Domain 2:
- 13 Non-participation,
- 15 Presence of non-participants,
- 16 Description of sample,
- 17 Pilot-tested?,
- 18 Repeat interviews,
- 21 Duration of interviews,
- 22 Data saturation,
- 23 transcripts returned,
- Domain 3:
- 25 Description of the coding tree (=framework matrix), Is this addressed by the supplementary data – if so, please state this.
- 27 Software
Lines 103: missing reference
Comments on Results section:
Line 110 Please add missing reference
Line 151 – (McCormack, 2003, p. 204) This reference should be the same as the overall referencing style throughout the manuscript.
Lines 176 – Please add full stop at sentence end.
Regarding need for extensive editing of style:
Several of the quotations are lacking quotation marks (either entirely or partially), making the results difficult to read. That is, there is a lacking distinction between the participants’ words and the authors’ words. This must be clearly rectified if the manuscript is to be published. Italicising the quotations (as is common) would also enable for easier reading and hence, overall comprehension of the manuscript.
Author Response
|
1 (Intro/General) |
1 |
The aim and relative objectives need to be made much clearer. |
This section is now in the introduction. |
|
|
2 |
There is no clear justification for the choice of method. |
Please see page 2 for further clarification over the methods, which includes outlining Narrative Inquiry and the reasons for using this method. |
|
|
3 |
Lines 30-31: Reference 1 is missing. It is incorrect for the referencing to start with Reference 2. Please correct. |
The document referencing has been refreshed, and the first sentence now contains reference 1 (page 1). |
|
|
4 |
Lines 48 -49: Patient-centred and person-centred are not interchangeable terms. They are two distinct types of practice. Please clarify the difference between the two and explain which of the two practices is being used in the manuscript. |
The phrasing ‘person-centred’ has been added to ensure consistency throughout the transcript. |
|
|
5 |
Line 49 – where does the quote end? There is no quotation mark (inverted commas) to close the quote. |
Quote marks have been added to indicate end of quote. |
|
Methods |
6 |
There justification for choice of methodology should be clearer. |
Further information relating to the choice of methodology has been added to the manuscript on page 3. |
|
|
7 |
The methods section does not provide enough detail for the study to be replicated which is a general requirement of a methods description. |
The authors have now added further information relating to the study methods (please see responses to comment 9). |
|
|
8 |
Lines 75-76 states that the “managers of the service were involved in reviewing the information for the study”. Why was this so and how could this have impacted the study.
|
Further information as to why managers’ involvement in reviewing findings has been added on page 2. |
|
|
9 |
The authors refer to having used the COREQ checklist in lines 76-77 which is appropriate. The COREQ checklist provides a “checklist of items that should be included in reports of qualitative research”. However, several COREQ items have not been reported. These missing items may influence what the participants felt comfortable in disclosing during the interviews and hence may have significantly impacted the results. These missing items should be reported to ensure the quality and rigor of the qualitative research such as sincerity, self-reflexivity, credibility, transparency, as listed (per domain and by item number) below. If the item has not been addressed then this should be explained in the methods section (in accordance with the chosen methodology) or addressed in a limitations section. · Domain 1: · 2 Credentials, · 3 Occupation, · 4 Gender, · 5 Experience and training, · 6 Relationship established, · 7 Participant knowledge of the interviewer, · 8 interviewer characteristics, · Domain 2: · 13 Non-participation, · 15 Presence of non-participants, · 16 Description of sample, · 18 Repeat interviews, · 21 Duration of interviews, · 22 Data saturation, · 23 transcripts returned, · Domain 3: · 25 Description of the coding tree (=framework matrix), Is this addressed by the supplementary data – if so, please state this. · 27 Software |
The authors thank the reviewer for highlighting this. Further information has now been added and highlighted through comments between pages 2 and 3, to comply with the COREQ checklist.
We’ve added in all the rest of the COREQ checklist now (previously removed to shorten the word count!)
|
|
|
10 |
Lines 103: missing reference |
The reference has now been included on page 3. |
|
Results |
11 |
Comments on Results section: Line 110 Please add missing reference |
The reference has now been included on page 4. |
|
|
12 |
Line 151 – (McCormack, 2003, p. 204) This reference should be the same as the overall referencing style throughout the manuscript. |
The correct formatting for the reference has now been changed to remain consistent with the referencing style. |
|
|
13 |
Lines 176 – Please add full stop at sentence end. |
A full stop has now been added here, on page 5. |
|
|
14 |
Regarding need for extensive editing of style: Several of the quotations are lacking quotation marks (either entirely or partially), making the results difficult to read. That is, there is a lacking distinction between the participants’ words and the authors’ words. This must be clearly rectified if the manuscript is to be published. Italicising the quotations (as is common) would also enable for easier reading and hence, overall comprehension of the manuscript. |
Quotes have been edited to distinguish between the participants’ and authors’ words. Quotes over four lines long have been expressed as block quotes. See pages 4-7 for further clarification. |
Reviewer 2 Report
Thank you for the opportunity to review this manuscript. It presents the findings from an interesting study focusing on the delivery of person-centered care in care homes. I have a number of comments that I have broken down by section which are intended to help future readers fully grasp the intention of the authors.
Introduction:
Pg 1, line 33-36 - the sentence starting, "Whilst pharmacist-led medication review interventions..." is awkward and difficult to follow. Is there another way it can be constructed to make it more clear?
Pg 1, line 38-40 - the part of the sentence starting, "...the project being proposed is ..." should be moved to the end of the introduction section as it is the study objective and will be clearer to the readers in that location.
Pg 2, line 51 - The sentence starting, "NICE guidelines 2015..." should begin a new paragraph.
Pg 2, line 62-63 - Was MOCH team created in response to the previously mentioned guidelines? More information here will help readers further contextualize the study results.
Methods:
General comment - I saw no mention in the methods section of how the interview questions were created, nor did I see a definition of "person-centered care". Providing a more complete definition of what the authors understand "person-centered care" to be will help the reader draw the connections for themselves between the data provided and the interpretations of the quotations within each of the themes by the authors.
Pg 3, line 94 - suggest making the sentence starting, "Framework analysis typically..." the beginning of a new paragraph.
Results:
General comment - It would be very helpful to the reader to the lengthier quotes offset to a greater degree from the rest of the text and for quotation marks to be added to them. In a couple of the sections it was difficult to always determine where the quotation ended and the authors description began.
Pg 3, line 118-120 - Since this quotation isn't terribly long I would suggest adding it to the previous paragraph. It is generally recommended that quotations be offset in this fashion if they are longer than four typed lines.
Pg 4, line 166 - Suggest spelling out the full title of CQC for audiences who are not familiar with this term.
Discussion:
Pg 6, line 268-276 - suggest moving section starting with, "The clinical reasoning associated with..." through to the end of the paragraph.
Pg 6 (logic model discussion) - It is not clear where the logic model came from, or how it was developed. If it was something that existed prior to the study, perhaps it would be better to mention it in the methods section and present the connection to it within the results section.
Pg 8, line 283-288 - It is not clear how this discussion relates to the larger findings of this study. Can the authors provide a more detailed set of connections?
Author Response
|
2 (introduction) |
1 |
Pg 1, line 33-36 - the sentence starting, "Whilst pharmacist-led medication review interventions..." is awkward and difficult to follow. Is there another way it can be constructed to make it more clear? |
First paragraph has been revised and clarified |
|
|
2 |
Pg 1, line 38-40 - the part of the sentence starting, "...the project being proposed is ..." should be moved to the end of the introduction section as it is the study objective and will be clearer to the readers in that location.
|
As above, lines 30-40 have been re-written |
|
|
3 |
Pg 2, line 51 - The sentence starting, "NICE guidelines 2015..." should begin a new paragraph.
|
Agreed, this has been done |
|
|
4 |
Pg 2, line 62-63 - Was MOCH team created in response to the previously mentioned guidelines? More information here will help readers further contextualize the study results. |
Line 66-68 clarifies the pilot implementation |
|
Methods |
5 |
General comment - I saw no mention in the methods section of how the interview questions were created, nor did I see a definition of "person-centered care". Providing a more complete definition of what the authors understand "person-centered care" to be will help the reader draw the connections for themselves between the data provided and the interpretations of the quotations within each of the themes by the authors. |
Further information relating to the development of the interview guide is included on pages 2-3 of the manuscript. |
|
|
6 |
Pg 3, line 94 - suggest making the sentence starting, "Framework analysis typically..." the beginning of a new paragraph. |
Agreed and amended but line 128 |
|
Results |
7 |
General comment - It would be very helpful to the reader to the lengthier quotes offset to a greater degree from the rest of the text and for quotation marks to be added to them. In a couple of the sections it was difficult to always determine where the quotation ended and the authors description began. |
Quotes have been edited to distinguish between the participants’ and authors’ words. Quotes over four lines long have been expressed as block quotes. See pages 4-7 for further clarification. |
|
|
8 |
Pg 3, line 118-120 - Since this quotation isn't terribly long I would suggest adding it to the previous paragraph. It is generally recommended that quotations be offset in this fashion if they are longer than four typed lines. |
Please see above response to comment 7 – as suggested, all quotes four lines or above have been offset. |
|
|
9 |
Pg 4, line 166 - Suggest spelling out the full title of CQC for audiences who are not familiar with this term. |
The full title for CQC has now been added into the manuscript in brackets (page 5). |
|
Discussion |
10 |
Pg 6, line 268-276 - suggest moving section starting with, "The clinical reasoning associated with..." through to the end of the paragraph.
|
Agreed and moved to P7 line 328 onwards |
|
|
11 |
Pg 6 (logic model discussion) - It is not clear where the logic model came from, or how it was developed. If it was something that existed prior to the study, perhaps it would be better to mention it in the methods section and present the connection to it within the results section.
|
P7 line 331 onward clarifies the data synthesis and a further section is in methods – with the Logic Model removed in favour of a more generic explanation see page 4 149-151 |
|
|
12 |
Pg 8, line 283-288 - It is not clear how this discussion relates to the larger findings of this study. Can the authors provide a more detailed set of connections? |
See p10 line 345 onwards “The evaluation sought to enable the service provider to select and share the most important aspects of the service and the contribution to person- centered care outcomes within the pilot… to 356 |
Reviewer 3 Report
The topic of this manuscript is of especial interest as is widely know that there is a great problem in long term care institutions in relation to the polypharmacy and the drug related problems derived from it.
The impact that the intervention of pharmacists and pharmacy technicians can make a great difference in the patient’s outcome.
The work is well organized and I think that is especially useful Table 1.
However I think that the manuscript could be improved in some points:
- As the Medicines Optimization in Care Homes teams is an initiative that is circumscribed to some parts of UK, It would be interesting to have a more detailed description of the components of the teams and their tasks in the everyday panorama. How do the team work?
- Methods: Even though that the authors indicate the general profile of the professionals doing the interview, it would be interesting to know, exactly what was the professional profile of the ones that made the 9 interviewees.
- Results: Did the authors found any difference between the answers of professionals of residences and nursing homes?
Author Response
|
1 |
As the Medicines Optimization in Care Homes teams is an initiative that is circumscribed to some parts of UK, It would be interesting to have a more detailed description of the components of the teams and their tasks in the everyday panorama. How do the team work?
|
Further information relating to the background of the participants has been added on page 2-3. |
|
2 |
Methods: Even though that the authors indicate the general profile of the professionals doing the interview, it would be interesting to know, exactly what was the professional profile of the ones that made the 9 interviewees.
|
Further information relating to the profile of the lead interviewer has been added on page 2 of the manuscript. |
|
3 |
Results: Did the authors found any difference between the answers of professionals of residences and nursing homes? |
We didn’t look at this within the analysis: but would hope to have the opportunity to re-analyse the data in future. |
Round 2
Reviewer 1 Report
The manuscript is significantly improved and is now easier to read. However, there are still some points to address.
Line 86 the word prescription seems inappropriate in the context. Suggest changing the word to prescribing or prescriptions as appropriate.
Inconsistency in spelling of person-centred. Original version used British spelling of person-centred whilst revised version tends to use American spelling of person-centered (lines 85, 92, 223, 463).
Add missing reference to NICE Guidance 2015 lines 73-74. Additionally, the same reference should be added to the following sentences that refer to the same publication (lines 76, 79, 81, 83).
Line 84: For consistency, change care to Care.
Inconsistency in reference formatting – lines 95 and 97-98, line 143
Line 97: add word ‘to’ after the word ‘and’ for better flow, that is, and to “adopt…
Inconsistency in British spelling. Line 111 change ‘behavorial’ to behavioural
Line 146: Suggest replacing “if the participant asked for more detail” with ‘when necessary’ for better flow.
COREQ Point 16 Suggest adding more detail at group level, including range and mean or median (depending on data skewness) of how many years’ experience, age variation as this may have impacted the results and will give the reader a deeper understanding of the participants.
COREQ Point 17 Not yet addressed. Please add.
COREQ Point 21 Please add mean.
COREQ Point 22 Lines 172-174 Data saturation should be reached and not decided upon. It is perfectly acceptable that the COVID-19 pandemic interrupted the data collection and hence data saturation could not be totally reached. However, can this be worded more appropriately?, such as although total data saturation was not entirely reached (very few new phenomenon, if any), the research team decided on adequate data saturation (ie no significant themes arising) at nine participants. Alternately, change the word ‘decided’ to the phrase ‘agreed upon’ if this was the case. This should be added as a limitation.
Line 176: Inconsistency in placement outside of quotation marks (compare with previous quote).
Line 177: Inconsistency in reference formatting
Comments on DISCUSSION section:
Generally, the argument as to how the findings connect person-centred care and medicines optimisation can be made stronger, that is, how the findings (of improved medicines optimisation) contribute to person-centred care, and to why this is beneficial (or not), needs to be more theorized/problematized to be more convincing.
Comparison of these findings to similar research is crucial yet lacking and needs to be addressed to give the findings a context.
Lines 301-303: Why is text italicised? Please correct.
Lines 324-326: Quotation lacking quotation marks.
Line 358: Please change conversation to plural form, i.e., conversations.
All quotations: please check for consistency in placement of full-stop.
Line 419: remove extra spacing. Suggest adding word ‘providers’ after ‘primary care’.
Line 423: The term ‘pharmaceutical care’ appears for the first time in the discussion without definition. Does this need to be further clarified to the readers? Are the readers aware of what pharmaceutical care is and how this is relevant to person-centred care or not?
Line 425: Please change “pharmacy review” to “medicines review”. Pharmacy review implies that an actual pharmacy was reviewed.
Lines 427-451 refer to results that are presented for the first time and hence should be presented in the results section. Please move to results section (including the table and reference to it). It is only appropriate to critique, discuss the implications of these findings or compare to other research in the discussion.
Line 463 Inconsistency in spelling: person-centered
Line 472-473 Should it be ‘health care teams’ or ‘healthcare teams’
Limitations not well reported.
Author Response
The manuscript is significantly improved and is now easier to read. However, there are still some points to address.
Line 86 the word prescription seems inappropriate in the context. Suggest changing the word to prescribing or prescriptions as appropriate.
I’m afraid that I’m struggling to find the word as line 86 on my copy has been deleted- perhaps this can be picked up in proofing publication?
Inconsistency in spelling of person-centred. Original version used British spelling of person-centred whilst revised version tends to use American spelling of person-centered (lines 85, 92, 223, 463).
I have changed these to the British spelling
Add missing reference to NICE Guidance 2015 lines 73-74. Additionally, the same reference should be added to the following sentences that refer to the same publication (lines 76, 79, 81, 83).
I have added [ibid] on line 72
Line 84: For consistency, change care to Care.
I think this has been corrected in Integrated Care System
Inconsistency in reference formatting – lines 95 and 97-98, line 143
These references have been removed in the formatting of the paper and just need to be allocated a number and the reference re-instated
Line 97: add word ‘to’ after the word ‘and’ for better flow, that is, and to “adopt…
This has been added
Inconsistency in British spelling. Line 111 change ‘behavorial’ to behavioural
Corrected
Line 146: Suggest replacing “if the participant asked for more detail” with ‘when necessary’ for better flow.
We prefer to leave as written
COREQ Point 16 Suggest adding more detail at group level, including range and mean or median (depending on data skewness) of how many years’ experience, age variation as this may have impacted the results and will give the reader a deeper understanding of the participants.
We are not in the position to share this data in the paper
COREQ Point 17 Not yet addressed. Please add.
Line 146 alludes to the topic guide with reference to Appendix 1. The interview plan wasn’t piloted
COREQ Point 21 Please add mean.
Line 147 identified the range of interview lengths we don’t think a mean value would add anything
COREQ Point 22 Lines 172-174 Data saturation should be reached and not decided upon. It is perfectly acceptable that the COVID-19 pandemic interrupted the data collection and hence data saturation could not be totally reached. However, can this be worded more appropriately?, such as although total data saturation was not entirely reached (very few new phenomenon, if any), the research team decided on adequate data saturation (ie no significant themes arising) at nine participants. Alternately, change the word ‘decided’ to the phrase ‘agreed upon’ if this was the case. This should be added as a limitation.
Amended and added as a limitation line 415
Line 176: Inconsistency in placement outside of quotation marks (compare with previous quote).
Line 177: Inconsistency in reference formatting
These lines don’t appear to include quotation marks or formatting problem- are the line references different?
Comments on DISCUSSION section:
Generally, the argument as to how the findings connect person-centred care and medicines optimisation can be made stronger, that is, how the findings (of improved medicines optimisation) contribute to person-centred care, and to why this is beneficial (or not), needs to be more theorized/problematized to be more convincing.
Comparison of these findings to similar research is crucial yet lacking and needs to be addressed to give the findings a context.
I have added a short section 408 onwards but we feel that the strength of the discussion is linked to the findings. We hope this is sufficient.
Lines 301-303: Why is text italicised? Please correct.
Lines 324-326: Quotation lacking quotation marks.
Both of these will need to be checked on publication proof
Line 358: Please change conversation to plural form, i.e., conversations. We think this is the change on 275
All quotations: please check for consistency in placement of full-stop. Completed
Line 419: remove extra spacing. Suggest adding word ‘providers’ after ‘primary care’. Done
Line 423: The term ‘pharmaceutical care’ appears for the first time in the discussion without definition. Does this need to be further clarified to the readers? Are the readers aware of what pharmaceutical care is and how this is relevant to person-centred care or not?
Changed to the service on 370 to avoid confusion
Line 425: Please change “pharmacy review” to “medicines review”. Pharmacy review implies that an actual pharmacy was reviewed.
Changed this on page 375
Lines 427-451 refer to results that are presented for the first time and hence should be presented in the results section. Please move to results section (including the table and reference to it). It is only appropriate to critique, discuss the implications of these findings or compare to other research in the discussion.
Line 463 Inconsistency in spelling: person-centered corrected
Line 472-473 Should it be ‘health care teams’ or ‘healthcare teams’ We have consistently used the former but editorial decision could override this?
Limitations not well reported. The suggested addition has been included
Reviewer 2 Report
Thank you for your careful considerations of my previous comments. I have nothing further to add.
Author Response
Many thanks for the review